# Two-Phase Extraction Processes, Physicochemical Characteristics, and Autoxidation Inhibition of the Essential Oil Nanoemulsion of *Citrus reticulata* Blanco (Tangerine) Leaves

**DOI:** 10.3390/foods12010057

**Published:** 2022-12-22

**Authors:** Marwan M. A. Rashed, Ling You, Abduljalil D. S. Ghaleb, Yonghua Du

**Affiliations:** 1Key Laboratory of Fermentation Resource and Application in Sichuan Higher Education, Faculty of Agriculture, Forestry and Food Engineering, Yibin University, Yibin 644001, China; 2School of Biological and Food Engineering, Suzhou University, Bianhe Middle Road 49, Yongqiao, Suzhou 234000, China; 3Faculty of Applied and Medical Science, AL-Razi University, Al-Rebatt St., Sana’a 216923, Yemen

**Keywords:** *Citrus reticulata* Blanco, two-phase extraction processes, essential oil nanoemulsion, autoxidation inhibition, antioxidant activity, physicochemical characteristics, creaming index

## Abstract

Combined ultrasound–microwave techniques and pre-enzymatic treatment (hemicellulase and cellulase) enhance essential oil isolation from *Citrus reticulata* Blanco (tangerine) leaves (CrBL). Subsequently, synergistic effects of modified amorphous octenyl succinic anhydride starch (OSA-MS), almond oil, and high-energy microfluidics were studied in synergy with ultrasound techniques in the production of CrBL essential oil (CrBL-EO) nanoemulsion (CrBL-EONE). GC–MS was used to study the extraction technique. Dynamic light scattering (DLS) analysis was used with confocal laser scanning microscopy (CLSM) techniques to investigate the nanoemulsion matrices’ physical and chemical properties. The D-limonene nanoemulsion (D-LNE) reached the optimal size of droplets (65.3 ± 1.1 r.nm), polydispersity index (PDI) (0.167 ± 0.015), and *ζ*-potential (−41.0 ± 0.4 mV). Besides, the CrBL-EONE obtained the optimal size of droplets (86.5 ± 0.5 r.nm), PDI (0.182 ± 0.012), and *ζ*-potential (−40.4 ± 0.8 mV). All the nanoparticle treatments showed significant values in terms of the creaming index (CI%) and inhibition activity (IA%) in the *β*-carotene/linoleate system with a low degradation rate (DR). The current study’s findings showed that integrated ultrasound–microwave techniques and pre-enzymatic treatment could enhance the extraction efficiency of the CrBL-EO. In addition, OSA-MS and almond oil can also be employed to produce CrBL-EONE and D-LNE.

## 1. Introduction

The consumers’ demand for food products with high nutritional value and health safety is increasing, especially with the increasing development in global trade, particularly in the food product sectors. Such emerging trends have led the industry and trade sectors to rethink the selection of food additives and rationalize their use, as well as to the gradual shift in replacing synthetic additives with natural alternatives [1].

Since ancient times, plant-based natural products have been a substantial source of several food additives and pharmaceutical products. Bioactive phytochemicals and plant-based sources can be isolated from flowers (e.g., *Zanthoxylum schinifolium* [2]), leaves (e.g., *Lavandula pubescens* [3]), fruit (e.g., *Solanum lycocarpum* St. Hil. (Solanaceae) [4]), stems (e.g., *Warburgia ugandensis* [5]), and roots (e.g., ginger [6]). Most phytochemicals found in plant-based sources fall into the following biochemical categories: polyphenols, terpenes, alkaloids, and glycosides [7]. Bioactive compounds, particularly those that form the chemical composition of essential oils, have attracted great interest from researchers and producers alike due to the diversity of their biological activities.

Medicinal plant use is widespread worldwide because of their distinctive chemical components that provide several desired bioactive properties. Since ancient times, traditional medicine therapists have used essential oils to prevent, avoid, or treat several diseases and provide the health benefits required. With the development of scientific research and the technological revolution in various aspects, essential oils have become very important due to their chemical composition that is rich in biologically active compounds and their importance in the production of healthy and nutritious products [8], processed foods [9,10], the pharmaceutical industry [11,12,13], cosmetics [14,15,16], anxiety reduction [17], and fragrance applications [18], either directly or indirectly.

The *Citrus* genus is an essential member of the Rutaceae family and includes tangerines, oranges, lemons, calamansi, limes, kumquats, mandarins, pomelos, and grapefruits. *Citrus reticulata* Blanco (tangerine) is one of the most significant members of the *Citrus* genus, and its essential oil has high health and economic benefits [19]. *Citrus* essential oils are widely used as an additive in food products, either to add a distinctive flavor [20] or to extend the shelf life of food products for a longer period due to their bioactive properties, such as antioxidant and antimicrobial action [21,22]. However, using essential oils as a replacement for synthetic additives is still in the early phase due to concerns about the chemical composition of some essential oils that are still being researched and their safety being investigated. Some of these concerns and limitations are, but not limited to, the safety of the chemical composition and effective concentration of the essential oil. In addition to the possibility of using essential oils in nanosystems, which has proven effective in maintaining the bioactive properties of many types of essential oils, the costs and benefits of applying EOs and nanotechnology in the field of food processing are also of interest [8].

Several modern techniques have been employed together with traditional extraction techniques to extract essential oils from aromatic plants. Some of these techniques have proven successful, and others are still being tested. The successful extraction techniques include liquid–liquid extraction, dispersed liquid–liquid microextraction, solid-phase extraction, solid-phase microextraction, ultrasound-assisted extraction, and microwave-assisted extraction [23].

Studies investigating the possibility of employing nanoemulsions as an emerging technology still attract many scholars in the food industry’s research and development sector. The success of a nanosystem is based on the production of an optimal nanoemulsion that meets the desired functional properties to increase the stability of the essential oil’s components against oxidation as well as its aqueous phase solubility and maintain its aromatic properties [2,24]. Despite the lipophilic properties of essential oils, they vary greatly in their chemical composition, making it necessary to carefully select the type of wall material in nanosystems to obtain highly stable nanoemulsions. For instance, the dominant components of rose oils are alcohol monoterpenes with a concentration of more than 70%, and citronellol is the major component [25]. In contrast, the dominant component of *Citrus* essential oils in general, and of the *C. reticulata* Blanco (tangerine) leaf essential oil in particular (the essential oil under study), is a linear monoterpene with a concentration of more than 90%, and D-limonene is the major component. Similarly to other essential oils, tangerine oil has low chemical stability and poor solubility in water [26]. Modified amorphous octenyl succinic anhydride starch (OSA-MS) is one of the most commonly used polysaccharides as a wall material in the preparation of nanoemulsions due to its distinctive chemical structure consisting of an amphipathic molecule with hydrophobic octenyl succinate groups and hydrophilic and hydroxyl groups of native starch. Still, the success and effectiveness of OSA-MS use are governed by the compatibility between the chemical composition of OSA-MS with the chemical composition of the core materials, the biologically active substances [25,27].

The current study focused on using the extracted essential oil from *C. reticulata* Blanco (tangerine) leaves using ultrasound–microwave techniques and pre-enzymatic treatment to enhance a novel form of nanoemulsion. The innovative technique can improve the physicochemical properties (*ζ*-potential, polydispersity index, and droplet size distribution) of nanoemulsion-based *C. reticulata* Blanco essential oils (CrBL-EO) using almond oil as a carrier substance and OSA-MS as a biodegradable emulsifying agent and wall material.

## 2. Materials and Methods

### 2.1. Plant Specimen and Chemicals

Fresh tangerine (*C. reticulata* Blanco) leaves (CrBL) were obtained from a local market in Suzhou City (Suzhou, Anhui, China). Hemicellulase and cellulase from *Aspergillus niger*, *β*-carotene (≥95%), and Trolox were purchased from Sigma-Aldrich (Darmstadt, Germany). Linoleic acid (85%), 2,2′-azino-bis(3-ethylbenzothiazoline-6-sulfonic acid) (ABTS), 2,2-diphenyl-1-picrylhydrazyl (DPPH), *tert*-butylhydroquinone (TBHQ), and 2,6-bis(1,1-dimethylethyl)-4-methylphenol (BHT) were purchased from TCI EUROPE N.V. (Brussels, Belgium). Modified amorphous octenyl succinic anhydride starch (OSA-MS) Hi-cap 100 was purchased from National Starch and Chemical Co. (Bridgewater, NJ, USA). Tween-40 and Tween-80 were procured from Sinopharm Chemical Reagent Co. Ltd. (Shanghai, China). Deionized distilled water used to prepare the nanoemulsion was produced using a Milli-Q water purification system (Millipore Corp., Darmstadt, Germany). Dyes nile red and fluorescein isothiocyanate (FITC) were purchased from Sigma-Aldrich Co. (Deisenhofen, Germany). All the other reagents used were of analytical grade.

### 2.2. Preparation of Tangerine Leaves

Tangerine leaves (CrBL) were soaked in an aqueous NaCl solution (1%, w/v), then washed with running tap water and gently dried at 40 ± 1 °C in an air oven (DHG-9245A, Shanghai Yiheng Scientific Instrument Co., Ltd., Shanghai, China) for 72 h. Then, the dried leaves were ground using a high-speed grinder to obtain a fairly fine powder. The fine powder sample (CrBL.P) was kept at 4 ± 1 °C in polyethylene bags protected from light until the next step. The moisture content of the CrBL.P was measured to be 6.5%.

### 2.3. Two-Phase Pre-Extraction of CrBL Essential Oil (CrBL-EO)

Two-phase extraction processes using pre-enzymolysis (first phase) followed by pre-ultrasound homogenizer-assisted extraction using a JY98-III DN Ultrasound Homogenizer, FeiQi Industry & Trade Co., Ltd. (Nanjing, China) (second phase), were performed. According to our previous study [23], with a slight modification, each fine powder sample of CrBL.P (100 g) was added to 500 mL of deionized distilled water that contained a mixture of 20 mg cellulase/hemicellulase at a ratio of 1:1 (w/w). The treatments were kept at 42 ± 1 °C for two h using a shaking water bath. All the treatments were subjected to the pre-ultrasound homogenizer-assisted phase. The ultrasound power and frequency of the pre-ultrasound homogenizer-assisted extraction were set at 150 W and 20 kHz. The total time programmed for the sonication treatment was 12 min, and the mixture was exposed to ultrasound waves for 5 min. The processing temperature (40 ± 2 °C) was continuously controlled using an external thermostatic cold-water bath. Each treatment was divided into two equal parts. Each treatment (in its two parts) was subjected to hydrodistillation for 30 min (for the first part) and 180 min (for the second part) using a Clevenger-type apparatus to isolate CrBL-EO. The hydrodistillation temperature was set at 100 ± 2 °C. [28]. The CrBL-EO samples obtained from each treatment were collected, weighed, and dried using anhydrous sodium sulphate. The collected samples were stored at 0 °C in brownish, airtight, and sealed vials. The treatments were coded as shown in Table 1.

The CrBL-EO yields (%) and the increase rates (IR) of the EOs were calculated according to Equations (1) and (2):(1)$$\mathrm{CrBL}-\mathrm{EO}\;Yield\%=\left[\frac{\mathrm{EO}\;\mathrm{obtained}\;\left(g\right)}{\mathrm{Initial}\;\mathrm{wt}.\;\mathrm{of}\;\mathrm{CrBL}\;\mathrm{powder}\;\left(g\right)}\right]\times 100$$
(2)$$\mathrm{IR}\%=\left[\left(\frac{A1}{A0}\right)\times 100\right]-100\;$$
where IR% is the increase rate, *A*1 is the EO obtained from a CrBL treatment, and *A*0 is the EO obtained from the control treatment (CrBL-EOc).

The separated CrBL-EO samples were kept in sealed brown glass vials at 4 °C until the next analysis.

### 2.4. Gas Chromatography–Mass Spectrometry (GC–MS) Analysis

The CrBL-EO treatments were analyzed to investigate the volatile compounds using an Agilent GC–MS system equipped with a gas chromatograph (7890A) and a mass spectrometer detector (5975C). The GC was fitted with an HP-5MS column (30 m × 0.25 mm internal diameter and 0.25 µm film thickness, J&W Scientific, Folsom, CA, USA) to separate the volatile compounds. Helium served as a carrier gas at a 1 mL/min flow rate with a 10:1 split ratio and a splitless injection volume of 1 µL. The gradient temperature was programmed as follows: 40 °C for 1 min, rising to 150 °C at 4 °C/min and maintained for 6 min; rising at 4 °C/min to 210 °C and maintained for 1 min. The injector and the detector were kept at 280 °C and 220 °C, respectively. For mass spectrometry (MS), electron ionization (EI) with an ionization energy voltage of 70 eV, emission current of 35 mA, and a spectral zone of 40–550 m/z was performed. The National Institute of Standards and Technology (NIST) 08, 2005 Library (Gaithersburg, MD, USA), Wiley 7 Library (Wiley, New York, NY, USA), and published mass spectra were used to identify the essential oil components based on their MS data.

### 2.5. Microstructure Observation Using Scanning Electron Microscopy (SEM)

The treatments (CrBL-EO_C2_, CrBL-EO_E2_, CrBL-EO_US2_, and CrBL-EO_EUS2_) were scanned using a SEM SU 1510 from Hitachi High-Technologies Corp. (Tokyo, Japan) to study the effect of two-phase extraction on the microstructure of CrBL.P leaf cells. All the prepared samples were fixed onto the sample holder using aluminum tape. The fixed samples were sputtered with a thin layer of gold and palladium using an ionic sputter coater. High vacuum conditions at an accelerating beam voltage of 5 kV were used. The working distance was set at 12.5 mm with a magnification coefficient of 1200×.

### 2.6. Measurement of Anti-Free Radical Activities

#### 2.6.1. DPPH^•^ Scavenging Activity Assay

Free radical scavenging activities of the CrBL-EO treatments were assessed as described previously [29] with a slight modification: 100 µL of each working sample at the final concentration of 0.2 mg/mL were mixed with 3 mL DPPH solution. The DPPH^•^ scavenging activity (DPPH^•^-SA%) was calculated as follows:(3)$${\mathrm{DPPH}}^{•}\text{-}\mathrm{SA}\%=\left[\frac{{OD}_{0}-{OD}_{1}}{{OD}_{0}}\right]\times 100$$
where *OD*_0_ and *OD*_1_ mean the optical density of the control sample and a targeted CrB-EO treatment.

#### 2.6.2. ABTS^•+^ Scavenging Activity (ABTS^•+^-SA) Assay

The ABTS^•+^-SA assay was performed according to Rashed et al. [23] with a slight modification: 100 µL of each working sample at the final concentration of 0.2 mg/mL were mixed with 3 mL ABTS solution. The calibration curve of Trolox was used to explain the free radical scavenging activity of the CrBL-EO treatments in mM Trolox.

Given that the CrBL-EO_EUS2_ treatment demonstrated the best yield (%), DPPH^•^-SE (%), and ABTS^•+^-SA (mM Trolox), it was selected for the rest of the subsequent analyses.

D-limonene, TBHQ, and BHT (100 µL each) were used as the positive samples, all at the same concentration (0.2 mg/mL).

### 2.7. Preparation of the CrBL-EO_EUS2_ Nanoemulsion

Nanoemulsions were produced in two phases: aqueous and organic. Exactly 3 g OSA-MS was dissolved with 92 mL deionized distilled water at 25 + 1 °C to prepare the aqueous phase. The solution was subjected to magnetic stirring overnight to ensure full hydration. The organic phase included 1.5 g almond oil, 2.5 g CrBL-EO_EUS2_, and 1 g Tween-80 (a nonionic surfactant). For full homogenization, the organic phase was subjected to magnetic stirring at 40 °C for 15 min and then at room temperature for 45 min (CrBL-EONE). A control sample (CNE) was prepared without adding CrBL-EO_EUS2_. A comparison sample was prepared using D-limonene (D-LNE) instead of CrBL-EO_EUS2_. Coarse emulsions were prepared by adding the organic phase drop-by-drop to the aqueous phase during the homogenization process at 18,000 rpm for 5 min using a high-speed homogenizer (Ultra Turrax T25 Basic, IKA, Staufen, Germany). The working treatments were then subjected to the pulsed sonication process at an ultrasound power of 960 W and the total sonication time of 2 s working time followed by 3 s elapsed time (total time of 15 min). All the nanoemulsions were subjected to further homogenization by passing through a high-pressure homogenizer at 100 MPa for five cycles using high-pressure homogenizer ATS AH2100 (ATS Engineering Inc., Brampton, ON, Canada). All the produced nanoemulsions were used in the subsequent analyses.

### 2.8. Physicochemical Characterization of the Nanoemulsion Treatments

The physicochemical properties of the CNE, CrBL-EONE, and D-LNE nanoemulsions were analyzed using a Zetasizer Nano (ZS, Malvern Instrument Ltd., Worcestershire, UK) based on the dynamic light scattering theory (DLS). The physicochemical assessment included the following measurements: *Z*-average in radius nanometers (r.nm), PDI of the size distribution, and *ζ*-potential. Each working sample (100 µL) was diluted 100× using deionized distilled water before analysis. All the measurements were performed at 25 °C in triplicate.

### 2.9. Creaming Index (CI%)

According to Rashed et al. [27], the creaming index (CI%) measurement was employed to evaluate the stability of the nanoemulsions prepared under storage at 25 °C for 30 days. The CI% was calculated as a percentage according to Equation (4):(4)$$\mathrm{CI}\%=\frac{TH_{NEL}}{TH_{SL}}$$
where *TH_NEL_* refers to the visual observation of the total initial layer height of the nanoemulsion, and *TH_SL_* is the separated serum layer height.

### 2.10. Confocal Laser Scanning Microscopy (CLSM)

The morphological investigation of the formulated nanoemulsions was conducted using a Zeiss Laser Scanning Microscopy 710 microscope (Zeiss Inc., Braunschweig, Germany) [30,31].

### 2.11. Thermal Autoxidation Using Inhibition of the β-Carotene/Linoleate System

Autoxidation inhibition of the CrBL-EO_EUS2_, D-limonene, CrBL-EONE, D-LNE, and TBHQ treatments was determined in the *β*-carotene/linoleic acid coupled system according to Rashed et al. [3] with a slight modification. To prepare the *β*-carotene/linoleate emulsion (*β*-CLE), 0.2 mg *β*-carotene was dissolved in 10 mL chloroform. Two mL of the *β*-CL solution were added into a round-bottom rotary flask with 20 mg linoleic acid and 200 mg Tween-40. After evaporating the chloroform using a rotary evaporator at 35 °C, 50 mL deionized distilled water (HPLC grade) was added to the reaction flask. The reaction substances were homogenized with manual shaking to obtain *β*-CLE. Five mL of *β*-CLE were pipetted into test tubes containing 0.2 mL (200 mgL^−1^) of the CrBL-EO_EUS2_, CrBL-EONE, or D-LNE treatments. In addition, 0.2 mL (200 mg/L) of standard TBHQ or D-limonene were used as the comparison samples. The control treatment consisted of 0.2 mL methanol instead of the extract. The zero-time absorbance was read at 470 r.nm once the emulsion was added to each test tube. All the samples were exposed to thermal autoxidation at 50 °C using a shaking water bath. Absorbance measurements were recorded every 20 min for 60 min. The disappearance of the β-carotene color in the control treatment indicates the end of the incubation time. All the treatments were analyzed in triplicate. The degradation rate (DR) of all the samples was calculated according to zero-order kinetic interactions according to the following Equation (5):(5)$$\mathrm{DR}=\frac{\ln\left(\frac{a}{b}\right)}{\frac{1}{60}}\;$$
where ln is the natural logarithm, *a* is the measured absorbance value at the zero time, and *b* refers to the measured absorbance value at the final time of treatment (60 min).

The percentage of the inhibition activity (IA%) was calculated as the inhibition ability of the CrBL-EO_EUS2_ treatment compared with the inhibition ability of the control sample according to the following Equation (6):(6)$$\mathrm{IA}\%=\left[\frac{DR_{0}-DR_{1}}{DR_{0}}\right]\times 100$$
where *DR*_0_ is the deterioration rate of the control treatment and *DR*_1_ is the target treatment’s (CrBL-EO_EUS2_, CrBL-EONE, D-LNE, TBHQ, and D-limonene) deterioration rate.

The differences between the IA% values at 20 min and 60 min (ΔIA%) were calculated (Equation (7)) to determine the stability of the emulsions during thermal treatment for 60 min.
(7)$$\Delta \mathrm{IA}\%={\;\mathrm{IA}\%}_{20\min}-{\mathrm{IA}\%}_{60\min}\;$$

### 2.12. Statistical Analysis

The data were expressed as the means ± standard deviation using SPSS 25.0 (SPSS Inc., Chicago, IL, USA). One-way analysis of variance (ANOVA) according to Duncan’s new multiple-range test was used to analyze the significant differences between the means. The differences were considered significant at *p* < 0.05.

## 3. Results

### 3.1. Two-Phase Extraction of CrBL-EO

Table 2 shows the yield (%) obtained for all the CrBL-EO treatments and the increase rate (IR%) between the 30 min hydrodistillation and the 180 min hydrodistillation. The yield (%) values from the highest to the lowest were as follows: 1.03, 0.9, 0.86, 0.86, 0.81, 0.8, 0.72, and 0.65% for CrBL-EO_EUS2_, CrBL-EO_EUS1_, CrBL-EO_US2_, CrBL-EO_E2_, CrBL-EO_US1_, CrBL-EO_E1_, CrBL-EO_C2_, and CrBL-EO_C1_, respectively. The yields of extracted essential oils are generally affected by several factors, the most important of which are origin, season, environmental factors, maturity stage (immature, semi-ripe, or mature), isolation technique, and isolation time [32,33]. Several studies focused on isolating the essential oil from *C. reticulata* Blanco (tangerine) waste. Hydrodistillation of *Citrus* peel from four *Citrus* species (*Citrus aurantifolia* (lime), *Citrus limon* (lemon), *Citrus limetta* (sweet lime), and *Citrus reticulata* (mandarin)) yielded an essential oil in the range of 0.48–0.75% after 4–5 h of hydrodistillation [34]. The present study obtained higher yield values (%) for *C. reticulata* Blanco (tangerine) leaves, with the maximum yield (%) of 1.03 ± 0.06 for the CrBL-EO_EUS2_ treatment; the lowest yield (%) was 0.65 ± 0.02 for the CrBL-EO_C1_ treatment. The isolation time during the hydrodistillation process significantly affected the yield increase (YI%) for all the treatments, with a YI% of 14.5% between the CrBL-EO_EUS1_ and CrBL-EOE_US2_ treatments, 11.86% between the CrBL-EO_C1_ and CrBL-EO_C2_ treatments, 7.47% between the CrBL-EO_E2_ and CrBL-EO_US1_ treatments, and 7.07% between the CrBL-EO_US2_ and CrBL-EO_EUS1_ treatments. The importance of enzymatic treatment is in its ability to dismantle the cellular structure of the cell wall in the plant tissue and depolymerize the polysaccharides of plant cell walls [23,35,36,37,38,39]. Hence, it would significantly increase the effectiveness of the subsequent ultrasound–microwave integrated treatment [29], thus enhancing the qualitative and quantitative (excluding yield%) quality of isolated CrBL-EO. Literature also reports that the cellulase achieved the highest yield of oil release from pumpkin seeds compared with the control sample using a microwave-assisted aqueous enzymatic extraction technique [40,41]. In general, these results illustrate that the mixture of hydrolysis enzymes (cellulase/hemicellulase) combined with ultrasound–microwave integrated treatment could enhance the release of the EO extracted from CrBL.P (CrBL-EO).

### 3.2. Chemical Composition of the CrBL-EO Treatments

Over 50 volatile components were identified in the CrBL-EO treatments. Table 2 shows the 23 volatile components at a concentration ≥ 0.5%. D-limonene was the main component of the CrBL-EO chemical composition in all the treatments. The concentrations (%) of D-limonene were as follows: 65.7, 59.35, 58.74, 57.2, 56.68, 54.14, 52.44, and 50.32% for EUS_30_ (CrBL-EO_EUS1_), C_30_ (CrBL-EO_C1_), E_30_ (CrBL-EO_E1_), US_30_ (CrBL-EO_US1_), EUS_180_ (CrBL-EO_EUS1_), E_180_ (CrBL-EO_E2_), US_180_ (CrBL-EO_US2_), and C_180_ (CrBL-EO_C2_), respectively. The results show that long-term hydrodistillation negatively affected the concentration of D-limonene in all the CrBL-EO treatments. According to its chemical properties, D-limonene is a natural cyclic monoterpene sensitive to exposure to high temperatures and light [33]. The chemical composition of a broad spectrum of essential oils isolated from most *Citrus* peels or leaves contains D-limonene [19,42]. Treatments with low processing time are compatible with two-phase extraction processes with high efficiency in enhancing the qualitative (antioxidant activity) and quantitative parameters (D-limonene%) of CrBL-EO treatments. This is because high-temperature treatment for a long time can lead to cracking the chemical composition of many of the main components of the EO, especially D-limonene [33]. Based on the literature, the results of this study appear to be consistent with the previously reported data [19].

C_30_ (CrBL-EO_C1_) and C_180_ (CrBL-EO_C2_) were the control treatments after 30 and 180 min of incubation time, respectively; E_30_ (CrBL-EO_E1_) and E_180_ (CrBL-EO_E2_) were the pre-enzymolysis treatments after 30 and 180 min of incubation time, respectively; US_30_ (CrBL-EO_US1_) and US_180_ (CrBL-EO_US2_) were the ultrasound–microwave treatments after 30 and 180 min of incubation time, respectively; EUS_30_ (CrBL-EO_EUS1_) and EUS_180_ (CrBL-EO_EUS1_) were the pre-enzymolysis and ultrasound–microwave treatments after 30 and 180 min of incubation time, respectively.

### 3.3. Microstructure Observation

The microstructure observation of the CrBL-EO_C2_ sample surface demonstrated slight ruptures with partial destruction after isolation of CrBL-EO without using any enzymatic or ultrasound–microwave treatments (Figure 1a) compared to the microstructure of CrBL-EO_E2_ (Figure 1b), CrBL-EO_US2_ (Figure 1c), and CrBL-EO_EUS2_ (Figure 1d). However, significant morphological and structural changes were observed in an enzymatic treatment (CrBL-EO_E2_), where most glandular structures were ruptured. These findings demonstrate that the destruction of cell walls when using hydrolysis enzymes such as cellulase and hemicellulase occurred and enhanced the release of CrBL-EO from the oily glands of plant cells. This might be attributed to the ability of cellulase and hemicellulase to decompose and hydrolyze the cellulose and hemicellulose layers in plant cell walls [38,43]. In the same context, the ultrasound–microwave treatments of CrBL-EO altered the surface microstructure of CrBL.P, as clearly demonstrated in Figure 1c,d. This effect can be attributed to the cavitation phenomenon resulting from ultrasound power combined with microwave radiation [44]. Therefore, the importance of employing a combination of hydrolysis enzymes (cellulase and hemicellulase) comes from the need to decompose the cell wall’s physical barrier to reach the storage glands of volatile and biologically active components and thus enhance the effectiveness of the ultrasound–microwave extraction which ultimately facilitates the access of the solvent into the glandular structures and secretory cavities [29,45].

### 3.4. Anti-Free Radical Activities

#### 3.4.1. DPPH^•^ Scavenging Activity (DPPH^•^-SA%)

The anti-free radical activity of the CrBL-EO treatments was measured using an in vitro DPPH^•^ assay. Table 3 shows the values of DPPH^•^-SA% of CrBL-EO_C1_, CrBL-EO_C2_, CrBL-EOE1, CrBL-EO_E2_, CrBL-EO_US1_, CrBL-EO_US2_, CrBL-EO_EUS1_, and CrBL-EO_EUS2_, D-limonene, TBHQ, and BHT. TBHQ (91.67%) and D-limonene (90.73%) had the highest value of DPPH^•^-SA%, with no significant differences, followed by CrBL-EO_EUS2_ (87.5%), CrBL-EONE (87.47%), CrBL-EO_EUS1_ (87.4%), BHT (85.97%), CrBL-EO_E2_ (82.93%), CrBL-EO_US2_ (82.83%), CrBL-EO_US1_ (82.8%), CrBL-EO_E1_ (82.7%), CrBL-EO_C2_ (81.37%), and CrBL-EO_C1_ (80.7%).

The increase rate (IR%) was calculated as a relative ratio attributed to the DPPH^•^-SA% value of the CrBL-EO_C2_ treatment (81.37%). The IR% values from the highest to lowest were as follows: 12.66, 11.51, 7.54, 7.5, 7.41, 5.65, 1.93, 1.8, 1.76, and 1.64% referring to TBHQ, D-limonene, CrBL-EO_EUS2_, CrBL-EONE, CrBL-EO_EUS1_, BHT, CrBL-EO_E2_, CrBL-EO_US2_, and CrBL-EO_US1_, respectively. Among all the CrBL-EO treatments, the CrBL-EO_EUS2_ treatment had the highest DPPH^•^-SA% value (87.5%).

#### 3.4.2. ABTS^•+^ Scavenging Activity (ABTS^•+^-SA)

As depicted in Table 3, the ABTS^•+^-SA values (mM Trolox) of the CrBL-EO treatments and positive treatments agreed and were highly consistent with the results obtained from the DPPH^•^-SA measurement. In this context, the ABTS^•+^-SA values (mM Trolox) of TBHQ (0.0360 ± 0.0012) with IR% of 96.36% followed by D-limonene (0.0330 ± 0.0014) with IR% of 80% were the highest among all the treatments. Among the CrBL-EO treatments, the CrBL-EO_EUS2_ treatment had the highest ABTS^•+^-SA value (0.0283 ± 0.0008) with an IR% of 54.55%, followed by CrBL-EO_EUS1_ (0.0280 ± 0.0007, IR% of 52.73) and CrBL-EONE (0.0277 ± 0.0004, IR% = 50.91). The ABTS^•+^-SA values (mM Trolox) of the control samples (CrBL-EO_C1_ and CrBL-EO_C2_) were 0.0173 ± 0.0011 and 0.0183 ± 0.0011, respectively.

The current study’s findings agree with the previous studies that described the ABTS assay (at 730 nm) as based on the generation of blue/green ABTS free radicals (ABTS^•+^) and as the optimal method for application in hydrophilic and lipophilic antioxidant systems. In contrast, DPPH free radicals in a DPPH^•^ organic solution are more suitable for application in hydrophobic systems [46].

### 3.5. Physicochemical Characterization of the CrBL-EONE Treatments

The frequency curves of the particle size distribution (*Z*-average) in r.nm, including the oversize, PDI, and *ζ*-potential (mV) curves of the CNE (control treatment), CrBL-EONE, and D-LNE treatments, are shown in Figure 2. The values of *Z*-average, PDI, and *ζ*-potential of the CNE treatment were 151.1 ± 6.0 r.nm, 0.356 ± 0.018, and –40.2 ± 0.6 mV, respectively. The values of *Z*-average, PDI, and *ζ*-potential for the CrBL-EONE treatment were 86.5 ± 0.5 r.nm, 0.182 ± 0.012, and –40.4 ± 0.8 mV, respectively. The values of the same parameters (*Z*-average, PDI, and *ζ*-potential) obtained by the D-LNE treatment were as follows: 65.3 ± 1.1 r.nm, 0.167 ± 0.015, and –41.0 ± 0.4, respectively. Several studies reported that droplet size, polydispersity index, and *ζ*-potential are among the most traceable physicochemical properties in colloidal systems to measure the stability of nanoemulsions and the integration of their components with high efficiency [8]. The obtained results demonstrated that the produced nanoemulsion-based CrBL-EO, almond oil as the carrier material, OSA-MS as the wall, and Tween-80 as the emulsification agent in synergy with the technique used to prepare the nanoemulsions all contributed to the production of nano-sized particles with homogeneous distribution in the nanoemulsion [27,47].

### 3.6. Creaming Index (CI%)

Figure 3 shows the creaming index (CI%) values of the nanoemulsion treatments, including CNE, CrBL-EONE, and D-LNE, after eight weeks of storage at 5 °C. There was no visually noticeable separation of the organic phase from the aqueous phase during the first three weeks, except for the CNE treatment, where the separation was exceedingly slight. Phase separation between the organic and aqueous phases started for all the treatments from the fourth week, although it was not significant as the CI% was 0.2, 0.1, and 0.1 for CNE, CrBL-EONE, and D-LNE, respectively. The CI% values remained almost stable during the fourth and fifth weeks. The values of the CI% increased from 0.4 to 0.6 and then 1.4 during the sixth, seventh, and eighth weeks, respectively.

On the other hand, the increase in the CI% was significantly lower compared with the CrBL-EONE and D-LNE treatments, which showed the same performance during the same storage time with the values of 0.2, 0.3, and 0.3, respectively. In general, the performance of the three treatments was acceptable and stable during the storage period, which lasted eight weeks at 5 °C. However, the CrBL-EONE and D-LNE treatments showed better performance and stability than the CNE treatment. Previous studies indicated several factors that may affect physicochemical properties, including the physical stability of essential oil nanoemulsions during storage. Such factors as nanoemulsion ingredients and preparation methods directly affect the values of particle size (*Z*-average), polydispersity index (PDI), and the droplets’ hydrostatic potential (*ζ*-potential) [48]. Particularly, the optimal values of *ζ*-potential (–20) are among the significant parameters that indicate the physical stability of nanoemulsions. Higher values of *ζ*-potential (negative or positive) lead to an increase in repulsive forces at the expense of attractive forces, which in turn leads to a decrease in the coalescence and creaming index values of nanoemulsion droplets [49], thus increasing the physical stability. The method of preparing the nanoemulsion that was applied to all the treatments under study based on high-speed homogenization followed by pulsed sonication and high-pressure homogenizer processes along with the effect of Tween-80 as a nonionic surfactant effectively contributed to achieving the optimal values of *ζ*-potential, which enhanced the physical stability of each of the three nanoemulsion treatments (Figure 2).

### 3.7. Microscopy CLSM Observation

Morphological observations using CLSM showed that the produced nanoemulsion droplets possessed a spherical and symmetrical shape, as shown in Figure 4a–c. The CLSM observations showed that OSA-MS efficiently coated the organic phase of CrBL-EO carried by almond oil. Among the CLSM observations, it was shown that CrBL-EONE (Figure 4b) and D-LNE (Figure 4c) had an advantage in terms of particle size, shape, and thickness compared with the control treatment (Figure 4a). The organic phase was incorporated into nanoemulsion formulation systems, and a nanoemulsion with spherical morphology was produced.

Similar findings were reported by Wang et al. [50] who studied the effect of OSA-MS on the physicochemical characteristics of mint flavor nanoemulsion.

### 3.8. Thermal Autoxidation and Stability Measurement

Figure 5 shows the IA% values of the CrBL-EO_EUS2_, CrBL-EONE, D-LNE, TBHQ, and D-Limonene treatments in the coupled system of *β*-CLE with the incubation time of 20, 40, and 60 min at 50 °C. The IA% values at the final incubation time (60 min) decreased dose-dependently in the following order: TBHQ (95.95%), D-LNE (95.1%), D-limonene (94.38%), CrBL-EO_EUS2_ (87.12%), and CrBL-EONE (86.43%).

The bleaching rate of the coupled *β*-CLE system was measured using the colorimetric method based on the optical density (470 nm) during 60 min of the total incubation time (Figure 6). The bleaching rate curves show a decrease in the optical density (470 nm) values in the presence of CrBL-EO_EUS2_, CrBL-EONE, D-LNE, D-Limonene, and TBHQ of the coupled system of *β*-CLE. The curves in Figure 6 show that the control treatment was less resistant and efficient in maintaining the color of *β*-carotene in the coupled system of *β*-CLE.

In contrast, the TBHQ treatment followed by the D-LNE and D-limonene treatments featured the highest resistance to thermal oxidation at 50 °C and degradation conditions among all the treatments, and thus the highest ability to protect the color of *β*-carotene in the *β*-CLE system, with no statistically significant differences between them. At the same time, both the CrBL-EO_EUS2_ and CrBL-EONE treatments showed a high ability to maintain the color of *β*-carotene and protect the *β*-CLE system from oxidation, with no significant differences in the optical density (470 nm) values between the two treatments (CrBL-EO_EUS2_ and CrBL-EONE).

Table 4 shows the DR (between 0 and 60 min, at 50 °C) and the *R^2^* values (between 0 to 60 min) of the *β*-CLE coupled system in the presence of the CrBL-EO_EUS2_, CrBL-EONE, D-LNE, D-limonene, TBHQ, and control treatments.

The lower the DR, the higher the efficiency of maintaining the *β*-CLE coupled system from deterioration and oxidation. The DR at the final incubation time (60 min) along with the *R^2^* values of the TBHQ, D-LNE, D-limonene, CrBL-EO_EUS2_, CrBL-EONE, and control treatments were as follows: 0.003 (*R*^2^ = 0.9526), 0.0036 (*R*^2^ = 0.8280), 0.0041 (*R*^2^ = 0.9300), 0.0094 (*R*^2^ = 0.9895), 0.0099 (*R*^2^ = 0.9856), and 0.0544 (*R*^2^ = 0.7500), respectively. Compared with the control treatment, the TBHQ, D-limonene, CrBL-EO_EUS2_, and CrBL-EONE treatments showed high resistance against oxidation processes.

By comparing the results in Figure 5 and Figure 6 and Table 4, it can be concluded that the TBHQ treatment had the highest long-term stability and IA% and the lowest DR (during 60 min at 50 °C) against the oxidation process, followed by the D-LNE, D-limonene, CrBL-EO_EUS2_, and CrBL-EONE treatments. In addition, *β*-CLE has a higher ability to resist oxidative deterioration and maintains the color of *β*-carotene sequentially from degradation when the chemical composition of the sample added can donate hydrogen ions (H^+^) or prevent the reaction of reactive oxygen species such as peroxides (ROOR), hydroxyl radical (HO^•^), superoxide (O_2_^−^), singlet oxygen (O=O), or α-O [51]. The added sample donates hydrogen ions to prevent or delay the oxidation process in *β*-CLE. The inhibition activity (IA%) of CrBL-EO_EUS2_ D-LNE, and CrBL-EONE can be attributed to their chemical composition rich in chemical compounds that donate free hydrogen ions H^+^.

## 4. Conclusions

In conclusion, it can be stated that cellulose and hemicellulase can hydrolyze plant cell walls, enhancing the efficiency of the essential oil isolated from the leaves of *Citrus reticulata* Blanco (tangerine). All the treatments of *C. reticulata* essential oil showed high potential in terms of the free radical scavenging activity. There was a synergistic effect between the enzymatic treatment using a mixture of cellulase/hemicellulase and the ultrasound–microwave integrated treatment. There was an integral effect between amorphous OSA-MS and almond oil in enhancing the efficiency of the bioactive *C. reticulata* essential oil nanoemulsion. The selected components of the produced nanoemulsion matrix contributed to the achievement of the desired values for each *Z*-average, PDI, and *ζ*-potential. The essential oil of *C. reticulata* and the formulated nanoemulsion of *C. reticulata* leaf essential oil had the optimal values of anti-free radical activity based on the DPPH and ABTS assays. In addition, they also had the highest lipid maintenance capacity of linoleic acid and β-carotene against oxidative degradation. OSA-MS showed excellent characteristics as a wall material in preparing *C. reticulata* leaf essential oil nanoemulsions due to its distinctive chemical composition consisting of an amphipathic molecule with hydrophobic and hydrophilic groups. Almond oil was also shown to be effective as a carrier material in protecting *C. reticulata* leaf essential oil. The current findings open up promising avenues for potential uses of *C. reticulata* leaf nanoemulsion as an additive in food products as a safe alternative to synthetic additives.

## Figures and Tables

**Figure 1 foods-12-00057-f001:**
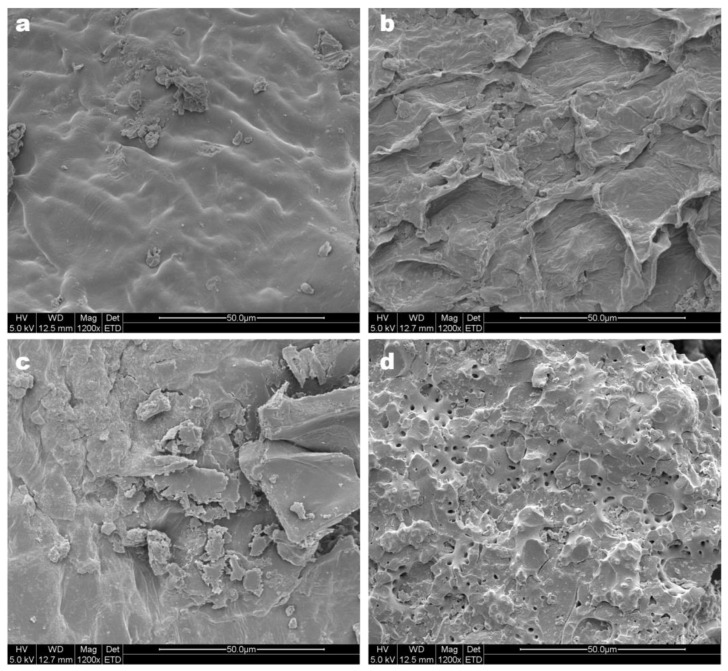
SEM observation of (**a**) the control treatment (CrBL-EO_C2_); (**b**) pre-enzymolysis treatment (CrBL-EO_E2_); (**c**) ultrasound–microwave treatment (CrBL-EO_US2_c); and (**d**) pre-enzymolysis and ultrasound–microwave treatment (CrBL-EO_EUS2_).

**Figure 2 foods-12-00057-f002:**
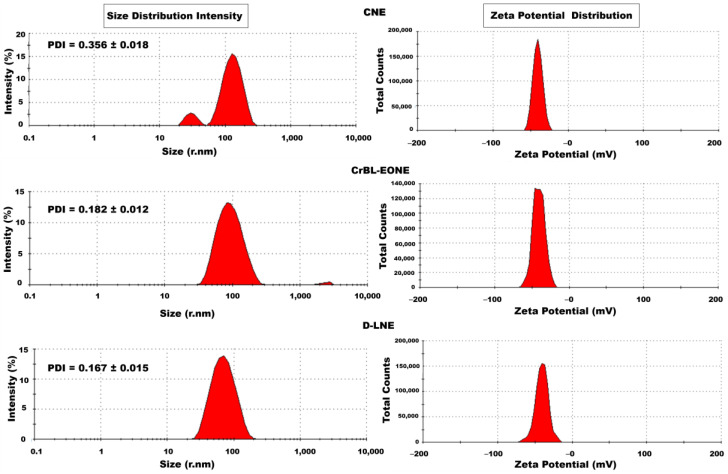
The frequency curves for *Z*-average include the oversize (logarithmic scale), PDI, and *ζ*-potential values of the control nanoemulsion treatment (CNE), *C. reticulata* leaf essential oil nanoemulsion (CrBL-EONE), and D-limonene nanoemulsion (D-LNE) (linear scale).

**Figure 3 foods-12-00057-f003:**
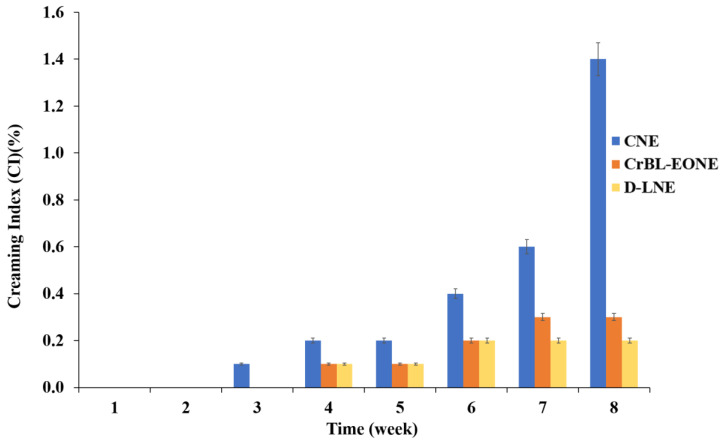
The creaming index (CI%) values of the control nanoemulsion treatment (CNE), *C. reticulata* leaf essential oil nanoemulsion (CrBL-EONE), and D-limonene nanoemulsion (D-LNE) after eight weeks of storage at 5 °C.

**Figure 4 foods-12-00057-f004:**
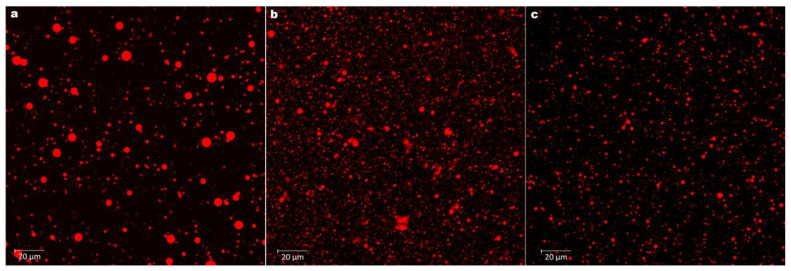
CLSM observations of (**a**) the control nanoemulsion (CNE) treatment, (**b**) *C. reticulata* leaf essential oil nanoemulsion (CrBL-EONE) treatment, and (**c**) D-limonene nanoemulsion (D-LNE) treatment.

**Figure 5 foods-12-00057-f005:**
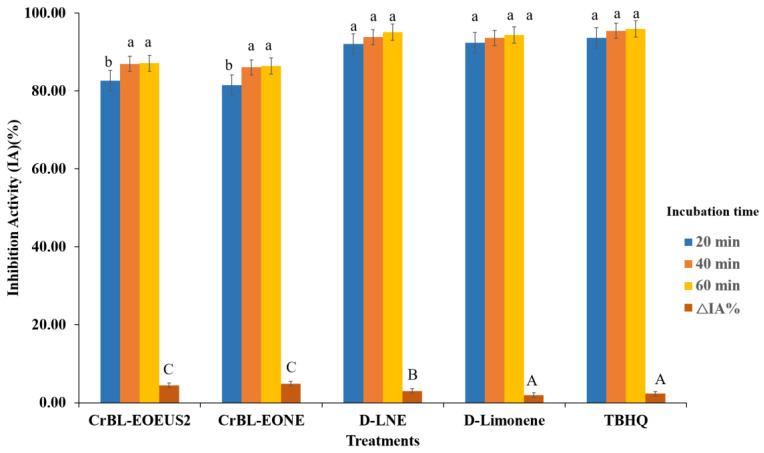
Inhibition activity (IA%) of the essential oil of *C. reticulata* with pre-enzymolysis and ultrasound–microwave treatment (CrBL-EO_EUS2_), *C. reticulata* essential oil nanoemulsion (CrBL-EONE), D-limonene nanoemulsion (D-LNE), D-limonene, and TBHQ in the *β*-carotene color in the coupled system (*β*-CLE). Bars bearing the same lowercase letters indicate no significant differences in the treatments during a unit of time (20, 40, 60 min). In comparison, bars bearing the same capital letters indicate no significant differences in the ΔIA% between the treatments.

**Figure 6 foods-12-00057-f006:**
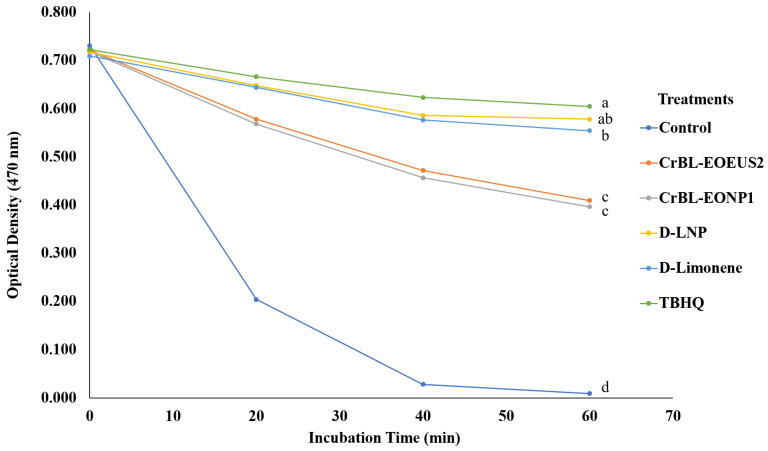
The bleaching rate curves (at an optical density of 470 nm) of the *β*-carotene color in the coupled system of *β*-CLE in the presence of the essential oil of *C. reticulata* with pre-enzymolysis and ultrasound–microwave treatment (CrBL-EO_EUS2_), *C. reticulata* essential oil nanoemulsion (CrBL-EONE), D-limonene nanoemulsion (D-LNE), D-limonene, and TBHQ. Lines with the same lowercase letters indicate no significant differences in treatment at the final incubation time (60 min).

**Table 1 foods-12-00057-t001:** Codes for the treatments subject to essential oil extraction.

Sample	Isolation Time	Code
CrBL.P without treatment (control)	30 min	CrBL-EO_C1_
	180 min	CrBL-EO_C2_
CrBL.P with pre-enzymolysis treatment	30 min	CrBL-EO_E1_
	180 min	CrBL-EO_E2_
CrBL.P with ultrasound–microwave treatment	30 min	CrBL-EO_US1_
	180 min	CrBL-EO_US2_
CrBL.P with pre-enzymolysis and ultrasound–microwave treatment	30 min	CrBL-EO_EUS1_
	180 min	CrBL-EO_EUS2_

**Table 2 foods-12-00057-t002:** Chemical composition of the CrBL-EO treatments.

RT ^1^	Components		Concentration (%)							
			C_30_	C_180_	E_30_	E_180_	US_30_	US_180_	EUS_30_	EUS_180_
01.352	Pentane		nd ^2^	2.00	nd	nd	nd	2.32	nd	nd
01.603	2-methylpentane		21.61	18.8	21.63	20.13	21.24	26.49	18.7	20.46
02.133	Cyclohexane		1.56	1.69	1.53	1.46	1.54	2.02	1.4	1.5
07.455	(−)-*α*-pinene		1.59	1.17	1.66	1.6	1.56	1.18	1.35	1.14
09.448	(−)-*β*-pinene		4.1	2.78	3.93	3.96	3.99	2.64	3.05	3.11
09.472	*β*-phellandrene		nd	tr ^3^	0.65	tr	0.84	tr	nd	tr
10.952	D-limonene		59.35	50.32	58.74	54.14	57.2	52.44	65.7	56.68
13.307	Linalool		0.94	tr	0.75	tr	1.05	tr	0.55	tr
14.857	*β*-terpineol		tr	tr	tr	0.55	tr	tr	tr	0.61
16.023	Terpinen-4-ol		1.76	1.19	1.14	1.08	1.29	0.97	1.25	1.02
16.530	*α*-terpineol		0.92	1.83	1.2	1.99	1.2	1.5	1.25	2.83
17.095	Decanal		0.83	0.62	0.99	0.8	1.29	0.78	0.75	0.91
17.900	*β*-citronellol		0.69	0.58	0.56	tr	0.71	0.5	0.75	0.62
19.392	dl-perillaldehyde		0.83	0.75	0.56	tr	0.61	0.5	0.85	0.79
19.998	*p*-mentha-1(7),8(10)-dien-9-ol		tr	tr	tr	tr	tr	nd	tr	0.87
20.785	2-methoxy-4-vinylphenol		nd	nd	nd	0.75	nd	nd	nd	0.35
23.373	(-)-cis-*β*-elemene		tr	tr	tr	0.56	tr	0.53	0.55	0.64
23.932	Dodecanal		tr	0.54	tr	0.72	tr	0.56	tr	0.83
26.159	(-)-germacrene D		tr	tr	tr	0.51	tr	tr	tr	0.62
31.836	Longifolene		nd	tr	nd	0.61	nd	nd	nd	0.5
36.592	*α*-sinensal		tr	1.87	0.55	1.84	tr	1.23	1.5	2.26
44.286	*n*-hexadecanoic acid		tr	2.28	tr	1.17	tr	0.53	tr	1.53
48.821	Linoleic acid		tr	0.72	tr	1.07	nd	nd	nd	nd
	Yield %	Total	0.65 ± 0.02 ^f^	0.72 ± 0.03 ^e^	0.80 ± 0.02 ^d^	0.86 ± 0.02 ^c^	0.81 ± 0.03 ^d^	0.86 ± 0.02 ^c^	0.90 ± 0.02 ^b^	1.03 ± 0.06 ^a^
		^4^ YI%	–	11.86	–	7.47	–	7.07	–	14.5

^1^ Retention time; ^2^ nd = not detected; ^3^ tr = trace; ^4^ YI—yield increase (%). Treatments with the means followed by the same superscripts in the columns did not differ significantly (*p* < 0.05).

**Table 3 foods-12-00057-t003:** DPPH^•^-SA (%) and ABTS^•+^-SA (mM Trolox) of the CrBL-EO treatments with IR% compared with the control treatment (CrBL-EO_C2_).

Treatment System	DPPH^•^-SA (%)		ABTS^•+^-SA (mM Trolox)	
	Result	IR ^1^	Result	IR
CrBL-EO_C1_	80.70 ± 0.20 ^e^	–	0.0173 ± 0.0011 ^e^	–
CrBL-EO_C2_	81.37 ± 0.42 ^e^	0.00	0.0183 ± 0.0011 ^e^	0.00
CrBL-EO_E1_	82.70 ± 0.20 ^d^	1.64	0.0213 ± 0.0011 ^d^	16.36
CrBL-EO_E2_	82.93 ± 0.15 ^d^	1.93	0.0217 ± 0.0011 ^d^	18.18
CrBL-EO_US1_	82.80 ± 0.10 ^d^	1.76	0.0210 ± 0.0012 ^d^	14.55
CrBL-EO_US2_	82.83 ± 0.25 ^d^	1.80	0.0220 ± 0.0007 ^d^	20.00
CrBL-EO_EUS1_	87.40 ± 0.30 ^b^	7.41	0.0280 ± 0.0007 ^b^	52.73
CrBL-EO_EUS2_	87.50 ± 0.30 ^b^	7.54	0.0283 ± 0.0008 ^b^	54.55
CrBL-EONE	87.47 ± 0.32 ^b^	7.50	0.0277 ± 0.0004 ^b^	50.91
D-limonene	90.73 ± 0.47 ^a^	11.51	0.0330 ± 0.0014 ^a^	80.00
TBHQ	91.67 ± 0.20 ^a^	12.66	0.0360 ± 0.0012 ^a^	96.36
BHT	85.97 ± 0.80 ^c^	5.65	0.0253 ± 0.0011 ^c^	38.18

^1^ IR—increase rate (%). The values were expressed as the means ± SD (*n* = 3). CrBL-EO_C1_ and CrBL-EO_C2_ were the control treatments after 30 and 180 min of incubation time, respectively; CrBL-EO_E1_ and CrBL-EO_E2_ were the pre-enzymolysis treatments after 30 and 180 min of incubation time, respectively; CrBL-EO_US1_ and US_180_ CrBL-EO_US2_ were the ultrasound–microwave treatments after 30 and 180 min of incubation time, respectively; CrBL-EO_EUS1_ and CrBL-EO_EUS1_ were the pre-enzymolysis and ultrasound–microwave treatments after 30 and 180 min of incubation time, respectively. Treatments with the means followed by the same superscripts in the columns did not differ significantly (*p* < 0.05).

**Table 4 foods-12-00057-t004:** The degradation rate (DR) and the *R^2^* values of the *β*-CLE coupled system in the presence of pre-enzymolysis and ultrasound–microwave treatment (CrBL-EO_EUS2_), *C. reticulata* leaf essential oil nanoemulsion (CrBL-EONE), D-limonene nanoemulsion (D-LNE), D-limonene, TBHQ, and control treatments between 0 min and the final incubation time (60 min) at 50 °C.

Treatment System	DR				
	0 min	20 min	40 min	60 min	*R* ^2^
Control	0	0.0213 ^c,A^	0.0544 ^c,B^	0.00544 ^c,B^	0.7500
CrBL-EO_EUS2_	0	0.0037 ^b,A^	0.0071 ^b,B^	0.0094 ^b,B^	0.9895
CrBL-EONE	0	0.0039 ^b,A^	0.0076 ^b,B^	0.0099 ^b,C^	0.9856
D-LNE	0	0.0017 ^a,A^	0.0034 ^a,B^	0.0036 ^a,B^	0.8280
D-limonene	0	0.0016 ^a,A^	0.0035 ^a,B^	0.0041 ^a,B^	0.9300
TBHQ	0	0.0014 ^a,A^	0.0025 ^a,B^	0.0030 ^a,B^	0.9526

The values were expressed as the means ± SD (*n* = 3). Small superscripts (in columns) indicate statistical significance between the treatments, while capital superscripts (in rows) indicate statistical significance between the incubation times. Treatments with the means followed by the same superscripts do not differ significantly (*p* < 0.05).

## Data Availability

Data is contained within the article.

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
