# Peer review of "Two-Phase Extraction Processes, Physicochemical Characteristics, and Autoxidation Inhibition of the Essential Oil Nanoemulsion of Citrus reticulata Blanco (Tangerine) Leaves"

_foods, 2022, doi:10.3390/foods12010057_

Round 1

Reviewer 1 Report

1.       Title. Please specify whether it was a nanoparticle or nanoemulsion.

2.       Line 24. Please delete Poly Dispersity Index, just say “PDI” because it is the second time.

3.       Line 28. Please state the type of enzyme used in this step.

4.       Line 43-51. Are there any drawbacks or limitations with using essential oils in food products? Also, keep it brief.

5.       Line 77. Please italicize Aspergillus niger

6.       Line 78. Please provide the city for Sigma-Aldrich (USA).

7.       Line 86. Wash with what solution?

8.       Line 89. What was the storage condition?

9.       Line 137-144. Since the ABTS test deals with polar free radicals as opposed to hydrophobic DPPH assay, please describe the technique in detail.

10.   How about the oxidative stability of the prepared nanoemulsion?

11.   Table 3. There is a very low yield. If it is intended to be scaled up industrially, kindly suggest a strategy to increase the yield.

12.   Table 3. Please specify the concentrations of D-limonene, TBHQ, and BHT used in the test. Also, the antioxidant activity of commercial tangerine essential oil at the same concentration should be compared.

13.   Throughout the manuscript, please use the number for temperature.

14.   Figure 2, please provide the bars of standard deviation and update the figure legend.

15.   Figure 4, please add the letters to indicate significant difference on the bars and update the figure legend.

16.   Please explain all abbreviations used in Table and Figure. To be understood, Table and Figure must be read independently.

17.   It is better to show the SEM microstructure of the leaves powder before and after enzymatic treatments to support the first sentence of Conclusion.

Author Response

Comment 1. Title. Please specify whether it was a nanoparticle or nanoemulsion.

Response: Done. "nanoparticles" has been replaced with "nanoemulsion". Please see page 1 (Line 4).

Comment 2. Line 24. Please delete Poly Dispersity Index, just say "PDI" because it is the second time.

Response: Done. Please see page 1 (Line 24).

Comment 3. Line 28. Please state the type of enzyme used in this step.

Response: Done. Please see page 1 (Lines 14 & 15).

Comment 4. Line 43-51. Are there any drawbacks or limitations with using essential oils in food products? Also, keep it brief.

Response: Done. Please see page 2 (Lines 59-68).

Comment 5. Line 77. Please italicize Aspergillus niger

Response: Done. Please see page 3 (Line 102).

Comment 6. Line 78. Please provide the city for Sigma-Aldrich (USA).

Response: Done. Please see page 3 (Line 107). Thanks for alerting us to this inadvertent typo.

Comment 7. Line 86. Wash with what solution?

Response: Edited. Please see page 3 (Lines 114 & 115).

Comment 8. Line 89. What was the storage condition?

Response: Edited. Please see page 3 (Line 117).

Comment 9. Line 137-144. Since the ABTS test deals with polar free radicals as opposed to hydrophobic DPPH assay, please describe the technique in detail.

Response: Clarification has been added. Please see page 9 (Lines 353-356).

Comment 10. How about the oxidative stability of the prepared nanoemulsion?

Response: The oxidative stability of the prepared nanoemulsion was already carried out in term of autoxidation inhibition using β-carotene/Linoleic acid coupled system. Please see the method section, pages 5 and 6 (Lines 219-247); results and discussions on pages 12-14 (Lines 431-491).

Comment 11. Table 3. There is a very low yield. If it is intended to be scaled up industrially, kindly suggest a strategy to increase the yield.

Response: We agree with the reviewer's comment regarding the yield (%) of Citrus peels, but when it comes to Citrus leaves, it is known that they contain much less essential oil than the yield (%) of essential oil in the peels. However, the extraction technique used in this study, based on ultrasound-microwave techniques and pre-enzymatic treatment, showed efficiency in enhancing the yield (%) of essential oil extracted from leaves, especially the CrBL-EOEUS2 treatment, as shown in Table 3.

This point is made clear in the manuscript text. Please see page 6 (Lines 259-268).

Comment 12. Table 3. Please specify the concentrations of D-limonene, TBHQ, and BHT used in the test. Also, the antioxidant activity of commercial tangerine essential oil at the same concentration should be compared.

Response: Done. Please see page 4 (Lines 183 & 184).

Clarification: C. reticulata leaf essential oil is rare and almost unavailable because most commercial types of citrus essential oils are those extracted from peels and not from the leaves. Our current study focused on the oil extracted from the leaves. For this reason, we used D-Limonene (a commercial standard compound) in comparison because it is the main constituent of most citrus essential oils, whether they are extracted from the leaves or peels, along with standard synthesis antioxidants (TBHQ and BHT).

Comment 13. Throughout the manuscript, please use the number for temperature.

Response: Done, and the entire manuscript has been checked.

Comment 14. In Figure 2, please provide the standard deviation bars and update the figure legend.

Response: Done. The numbering of Figure 2 has been changed to Figure 3. Please see Figure 3 after edit on page 11.

Comment 15. Figure 4, please add the letters to indicate significant difference on the bars and update the figure legend.

Response: Done. The numbering of Figure 4 has been changed to Figure 5. Please see Figure 5 after edit on page 12.

Comment 16. Please explain all abbreviations used in Table and Figure. To be understood, Table and Figure must be read independently.

Response: Done. The entire manuscript has been checked.

Comment 17. It is better to show the SEM microstructure of the leaves powder before and after enzymatic treatments to support the first sentence of Conclusion.

Response: Done. Please see Methods on page 4 (Lines 158-165), Results and Discussions on pages 7 and 8 (Lines 306-372), and Figure 1 on page 8.

All comments and inquiries raised by the reviewer have been handled thoroughly. The authors sincerely Thank and Appreciate the Reviewer for these Valuable Comments.

Reviewer 2 Report

Review report

Manuscript ID: foods-2043024 

Title: Two-phase extraction processes, physiochemical characteristics, 2 and autoxidation inhibition of Citrus reticulata Blanco (Tange-3 rine) leaves essential oil

 The authors considered the extraction of essential oil from C. reticulata Blanco leaves by means of a combined ultrasound-microwave technique and a pre-enzymatic treatment. The objective is to enhance a novel form of nanoemulsion. They claimed that the innovation technique could lead to an improvement of the techno-functional properties and sustainable applications of nanoparticles-based C. reticulata Blanco essential.

The topic is interesting and the paper well presented. However the authors should clarify the following points:

-          How they did set the operating conditions for the two-phase extraction process, particularly concerning temperature, material quantities and proportions, volumes, treatment time, etc;

-          What are the reasons that long-term hydrodistillation negatively affected the concentration of D-Limonene in all CrBL-EO treatments;

-          Why the phase separation between the organic and aqueous phases started for all treatments from the fourth week;

-          Why treatments with low processing time corresponded to Two-phase extraction processes with high efficiency in enhancing the qualitative and quantitative parameters of CrBL-EO treatments;

-          The paragraph “The importance of enzymatic treatment is in its ability to dismantle the cellular structure of the cell wall in the plant tissue and depolymerize the polysaccharide of plant cell walls [20, 28–30]. This would significantly increase the effectiveness of the subsequent ultrasound/microwave-integrated treatment [25], thus enhancing the qualitative and quantitative quality of isolated CrBL-EO” is repeated but just with different reference numbers;

-          Please correct Equation 2, it should be A1/A0 not A1/A1

-          English writing should be checked in some parts of the text. For instance in the paragraph “In general, these results illustrated that the mixture of hydrolysis enzymes (CE/HE) combined with ultrasound/microwave-integrated treatment could be enhanced the release of the EO extracted from CrBL (CrBL-EO).”

Finally I urge the authors to address the above raised points before any acceptance of the paper is envisaged

Author Response

Point 1. How they did set the operating conditions for the two-phase extraction process, particularly concerning temperature, material quantities and proportions, volumes, treatment time, etc.

Response: Done. All points raised by the reviewer are explained in the manuscript text. Please see page 3 (Lines 124-134).

Point 2. What are the reasons that long-term hydrodistillation negatively affected the concentration of D-Limonene in all CrBL-EO treatments?

Response: According to its chemical properties, D-Limonene is a natural cyclic monoterpene sensitive to high temperatures and light exposure.

- This justification has been added to the manuscript text. Please, see page 6 (lines 259-268) and page 7 (Lines 292-297).

Point 3. Why the phase separation between the organic and aqueous phases started for all treatments from the fourth week?

Response: The justification has been added to the manuscript text. Please, see page 11 (Lines 404-416).

Point 4. Why treatments with low processing time corresponded to Two-phase extraction processes with high efficiency in enhancing the qualitative and quantitative parameters of CrBL-EO treatments.

Response: A clarification of this inquiry has been added to the discussion part. Please see page 7 (Lines 292-297).

Point 5. The paragraph "The importance of enzymatic treatment is in its ability to dismantle the cellular structure of the cell wall in the plant tissue and depolymerize the polysaccharide of plant cell walls [20, 28–30]. This would significantly increase the effectiveness of the subsequent ultrasound/microwave-integrated treatment [25], thus enhancing the qualitative and quantitative quality of isolated CrBL-EO" is repeated but just with different reference numbers.

Response: Corrected. Thanks for that note. Duplicate paragraphs were removed and the manuscript was checked to avoid similar unintentional errors.

Point 6. Please correct Equation 2. It should be A1/A0, not A1/A1

Response: Corrected. Please see page 4 (Equation 2).

Point 7. English writing should be checked in some parts of the text. For instance in the paragraph "In general, these results illustrated that the mixture of hydrolysis enzymes (CE/HE) combined with ultrasound/microwave-integrated treatment could be enhanced the release of the EO extracted from CrBL (CrBL-EO)."

Response: Done. Thanks for that note. Duplicate paragraphs were removed, and the manuscript was checked to avoid similar unintentional errors.

Finally, I urge the authors to address the above raised points before any acceptance of the paper is envisaged.

Response: All comments and inquiries raised by the reviewer have been handled thoroughly.

The authors Sincerely Thank and Appreciate the Reviewer for these Valuable Comments.

Reviewer 3 Report

The paper has seriously flaws . The main drawback of the paper is the discussion part which can be considered missing since only a few references were stated and no profound explanations are present. The introduction is too general with no specific interest. Additionally no practical application was discussed in the paper which is very important part of scientific research.

Author Response

Point 1. The discussion part which can be considered missing since only a few references were stated and no profound explanations are present.

Response: The discussion section has been modified according to the reviewer's suggestion. Please see pages 6 (Lines 259-268), page 7 (Lines 289-297), Pages 7 and 8 (Lines 314-323), page 9 (Lines 353-356), page 10 (Lines 374-377), page 11 (Lines 404-416), and page 14 (Lines 485-489). In the Conclusion section: please see page 14 (Lines 504-507).

Point 2. The introduction is too general, with no specific interest.

Response: The introduction section has been modified. Please see pages 1 (Lines 34-38) and 2 (Lines 59-68, 74-92, and 95-96).

Point 3. Additionally no practical application was discussed in the paper which is very important part of scientific research.

Response:

         We understand and realize what the reviewer means. Still, we want to clarify that this research aims to study the effect of a two-stage extraction method on the qualitative and quantitative characteristics of Citrus reticulata Blanco (Tangerine) leaves essential oil and then enhance its stability during the storage time throughout applying a nanosystem based on highly efficient multi-stage homogenization techniques (The method of preparing the nanoemulsion that was applied to all the treatments under study, based on the high-speed homogenizer followed by pulsed sonication and high-pressure homogenizer processes along with the effect of Tween-80 as a non-ionic surfactant, all effectively contributed to achieving the optimal values of ζ-potential, which enhanced the physical stability of each of the nanoemulsion treatments). However, applying the current study findings is not oblivious to the research team, which is the aim of the following study we are conducting. We intend to publish its results in the Foods Journal as well. Still, we will not be able to include its results in the current study (Manuscript ID: foods-2043024) because it will include the investigation of various and multiple indicators and parameters. This includes microbiological and anti-enzyme studies, applications in baked products and meat to enhance the quality criteria, associated analytical indicators, sensory evaluation, etc. All these analyzes are ramified and broad, so it isn't easy to include them in the current study. In addition, All these analyzes are interrelated, so they cannot be fragmented because this would affect the interdependence of the analyses.

Comments and inquiries raised by the reviewer have been handled thoroughly.

The Authors Sincerely Thank and Appreciate the Reviewer for these Valuable Comments.

Reviewer 4 Report

I reviewed the manuscript entitled "Two-phase extraction processes, physiochemical characteristics, and autoxidation inhibition of Citrus reticulata Blanco (Tangerine) leaves essential oil nanoparticles" by Marwan M. A. Rashed at al. The authors described obtaining and characterization of nanoparticles isolated from Citrus reticulata Blanco (tangerine) leaves by combined ultrasound-microwave techniques and pre-enzymatic treatment.

In my opinion, this manuscript is not clear and well to understand. There are many abbreviations, along the manuscript, while other like ABTS, DPPH, TBHQ, and BHT (line 79), OSA-MS (line 147), CE/HE (line 229), ζ-pot are not described. Also, some materials such as α-tocopherol (line 78), lavender nanoemulsion (line 81) did not used in research methodology.

Table 1 contained data that are not explained (such as, isolation time).

The Pdi is not represented in Fig. 1, as the authors said. Some graphs are in logaritmic scale, not liniar.

Although the authors cited references mostly recent published, the obtained data for antioxidant activity, creaming index, thermal autoxidation and stability measurement are not commented according with literature.

What means in r.nm at Z-average (Z-ave)? For natural logaritm the abbreviation is ln not In.

The experimental results are not well correlated with the described methodology.  

What is the novelty of this research? OSA-modified starch was also used to protect bioactive compounds such as essential oils (Xiao, Z., Kang, Y., Hou, W., Niu, Y., & Kou, X. (2019). Microcapsules based on octenyl succinic anhydride (OSA)-modified starch and maltodextrins changing the composition and release property of rose essential oil. International Journal of Biological Macromolecules, 137, 132–138. doi:10.1016/j.ijbiomac.2019.06.17).

Based on the above comments I consider that this manuscript should be reconsider after major revision.

Author Response

Point 1. There are many abbreviations, along the manuscript,

Response: All of the following abbreviations have been omitted to reduce their number from the manuscript text:

  1. US-HA (was used to refer to pre-ultrasound homogenizer-assisted treatment);
  2. ase (was used to refer to Cellulase);
  3. ase (was used to refer to Hemicellulase);
  4. DDW (was used to refer to Deionized distilled water);
  5. Z-ave (was used to refer to Z-average);
  6. ζ-pot (was used to refer to ζ-potentials);
  7. OD470 (was used to refer to the optical density 470 nm)

Point 2. while other like ABTS, DPPH, TBHQ, and BHT (line 79), OSA-MS (line 147), CE/HE (line 229), ζ-pot are not described.

Response:

  • According to the reviewer's comment, the abbreviations (ABTS, DPPH, TBHQ, BHT, and OSA-MS) have been described. Please see page 3 (lines 103-108).
  • The following abbreviations have been replaced by their full names to reduce the abbreviation numbers according to comment#1:

CE/HE has been changed to Cellulase/Hemicellulase wherever it is found in the manuscript.

ζ-pot has been changed to ζ-potential wherever it is found in the manuscript.

We have checked that there were no similar mistakes throughout the manuscript.

Point 3. Also, some materials such as α-tocopherol (line 78), lavender nanoemulsion (line 81) did not used in research methodology.

Response: These two items were mistakenly added to the manuscript and they have been deleted.

Point 4. Table 1 contained data that are not explained (such as, isolation time).

Response: The isolation time was already indicated in Table 1, where the treatments subject to isolation for 30 min were given the number 1, and their codes are as follows: CrBL-EOC1, CrBL-EOE1, CrBL-EOUS1, and CrBL-EOEUS1. At the same time, treatments that were subject to isolation for 180 min were given the number 2, and their codes are as follows: CrBL-EOC2, CrBL-EOE2, CrBL-EOUS2, and CrBL-EOEUS2.

So, the reference to these treatments throughout the manuscript with these codes is originally a reference to the isolation time.

For more accuracy, the following explanations and clarifications have been added to the manuscript according to the reviewer's suggestion:

Method: please see page 3 (lines 128-134).

Results and Discussions: please see page 6 (lines: 256, 259-268), and page 7 (Lines 292-297).

Footnote to Table 2: please see page 7 (300-305).

Footnote to Table 3: please see page 9 (lines 360-365).

Point 5. The Pdi is not represented in Fig. 1, as the authors said. Some graphs are in logaritmic scale, not liniar.

Response: Done. The numbering of Figure 1 has been changed to Figure 2. Please see Figure 2 page 10.

Point 6. Although the authors cited references mostly recent published, the obtained data for antioxidant activity, creaming index, thermal autoxidation and stability measurement are not commented according with literature.

Response: Modified. Please see page 9 (lines 353-356), page 10 (lines 374-377), page 11 (404-416), and page 14 (lines 485-489).

Point 7. What means in r.nm at Z-average (Z-ave)?

Response: r.nm refers to the radius in nanometers. It has been added to the manuscript. Please see page 5 (line 205).

The Z-ave abbreviation (refers to Z-average). The Z-ave abbreviation has been deleted to reduce the number of abbreviations.

Point 8. For natural logaritm the abbreviation is ln not In.

Response: Done and modified.

Point 9. The experimental results are not well correlated with the described methodology.  

Response: Done. The methodology, results, and discussion sections have been updated to ensure their interrelationship.

Point 10. What is the novelty of this research? OSA-modified starch was also used to protect bioactive compounds such as essential oils (Xiao, Z., Kang, Y., Hou, W., Niu, Y., & Kou, X. (2019). Microcapsules based on octenyl succinic anhydride (OSA)-modified starch and maltodextrins changing the composition and release property of rose essential oil. International Journal of Biological Macromolecules, 137, 132–138. doi:10.1016/j.ijbiomac.2019.06.17).

Response: Of course, several studies have used amorphous octenyl succinic anhydride-modified starch (OSA-MS) as a wall material in nanoemulsion systems. Still, the success and effectiveness of its use are governed by the compatibility between the chemical composition of the OSA-MS with the chemical composition of the core materials, the biologically active substances. In this context, essential oils differ greatly in their chemical structures, making it necessary to carefully select the type of wall material in the nanosystem to obtain a highly stable nanoemulsion. For instance, the dominant composition of rose oils is alcohol monoterpene (Citronellol is the major component), with a concentration of more than 70%. In contrast, the dominant composition of Citrus essential oils in general, and C. reticulata Blanco (Tangerine) leaves in particular (the oil under study) is a linear monoterpene (D-Limonene is the major component) with a concentration of more than 90%. This is what was sought and confirmed by the results of the current study that there is an agreement between the chemical composition of the modified starch and the chemical composition of tangerine oil.

Action: This clarification has been added to the introduction and discussions with cited to the reference recommended by the reviewer. Please see page 2 (lines 86-92 & 95, 96), page 10 (lines 374-377), and page 11 (Lines 404-416).

  • Based on the above comments I consider that this manuscript should be reconsider after major revision.

Response: All comments and inquiries raised by the reviewer have been handled thoroughly.

The Authors Sincerely Thank and Appreciate the Reviewer for these Valuable Comments.

Round 2

Reviewer 1 Report

All points raised by reviewers were carefully addressed and answered point-by-point. So, it can be accepted.

Author Response

The authors thank the reviewer for his time and valuable comments.

Reviewer 3 Report

I agree with the authors that the results can not be fragmented but my point was to present the potential aplication to the readers. This should be added as few sentences in the conclusion part.

Author Response

Comment: I agree with the authors that the results can not be fragmented but my point was to present the potential aplication to the readers. This should be added as few sentences in the conclusion part.

Response: Done. Please see page 14 (lines 506-508).

The authors sincerely Thank and Appreciate the Reviewer for all Valuable Comments during peer-review processes (R1 and R2).

Reviewer 4 Report

I received the improved manuscript  according to the reviewers' suggestions.

However, there are still necessary some corrections to be made.  

1) I kindly suggest the authors to check the current number of figures and tables inside of manuscript (for ex. at line 253, Table 3 is mentioned before of Table 2; at line 315 appears Fig. 3c and 3d). Also, another Table 3 appears at Line 472.

2) Line 315: the abbreviation "CrBL-P" is not explained. Also, "NEs" from line 377. Which abbreviation is correct, the one from Line 214 or Line 415 (CLSM or CLMS)?

3) Line 380: "Figure 2. The frequency curves for Z-average include oversize" (in logarithmic scale) and PDI.....

4) Figure 3: Please complete the label of OY axis "Creaming Index (CI) (%)". Abscissa axis should contains 1...8, without "W". Similar, the OY axis from Fig. 5.

Author Response

Point 1.  I kindly suggest the authors check the current number of figures and tables inside of the manuscript (for ex. at line 253, Table 3 is mentioned before Table 2; at line 315 appears Fig. 3c and 3d). Also, another Table 3 appears at Line 472.

Response: Done.

  • Results belonging to Yield (%) and Yield Increase (YI%) have been moved from Table 3 to Table 2 to avoid confusion. Please see page 6 (line 253) and page 7 (Table 2).
  • Table number 3 has been corrected to Table 4. Please see page 13 (line 456) and page 14 (lines 478 and 481).
  • Figures 3c and 3d have been corrected to Figures 1c and 1d. Please see page 8 (316).
  • The entire manuscript has been

Point 2.  Line 315: the abbreviation "CrBL-P" is not explained. Also, "NEs" from line 377. Which abbreviation is correct, the one from Line 214 or Line 415 (CLSM or CLMS)?

Response: Done.

  • CrBL-P abbreviation has been corrected to CrBL.P. Please see page 8 (line 316). The first mention of CrBL.P is on page 3 (line 116).
  • NEs abbreviation has been corrected to nanoemulsions. Please see page 10 (line 78).
  • CLSM is the correct abbreviation. CLMS has been corrected to CLSM. Please see page 11 (line 416).

Point 3.   Line 380: "Figure 2. The frequency curves for Z-average include oversize" (in logarithmic scale) and PDI.....

Response: Done. It has been modified according to the comment. Please see page 10 (line 382).

Point 4.  Figure 3: Please complete the label of the OY axis "Creaming Index (CI) (%)". The abscissa axis should contain 1...8, without "W". Similarly, to the OY axis from Fig. 5.

Response: Done.

  • Please see Figure 3 (page 11) and Figure 5 (page 12).

All comments and inquiries raised by the reviewer have been addressed thoroughly.

The authors sincerely Thank and Appreciate the Reviewer for these Valuable Comments during peer-review processes (R1 and R2).